# Real-World Comparison of Trifluridine–Tipiracil with or Without Bevacizumab in Patients with Refractory Metastatic Colorectal Cancer

**DOI:** 10.3390/biomedicines13040976

**Published:** 2025-04-16

**Authors:** Hyunho Kim, Kabsoo Shin, Ho Jung An, In-Ho Kim, Jung Hoon Bae, Yoon Suk Lee, In Kyu Lee, MyungAh Lee, Se Jun Park

**Affiliations:** 1Division of Medical Oncology, Department of Internal Medicine, St. Vincent’s Hospital, College of Medicine, The Catholic University of Korea, Suwon 16247, Republic of Korea; hori1104@naver.com (H.K.);; 2Division of Medical Oncology, Department of Internal Medicine, Seoul St. Mary’s Hospital, College of Medicine, The Catholic University of Korea, Seoul 06591, Republic of Korea; 3Cancer Research Institute, College of Medicine, The Catholic University of Korea, Seoul 06591, Republic of Korea; 4Division of Colorectal Surgery, Department of Surgery, Seoul St. Mary’s Hospital, College of Medicine, The Catholic University of Korea, Seoul 06591, Republic of Korea; eysi0815@catholic.ac.kr (J.H.B.); cmcgslee@catholic.ac.kr (I.K.L.)

**Keywords:** colorectal cancer, trifluridine–tipiracil, bevacizumab, real-world evidence

## Abstract

**Background/Objectives:** Patients with metastatic colorectal cancer (mCRC) who are refractory to standard chemotherapy face limited treatment options. While trifluridine–tipiracil (FTD–TPI) and regorafenib have shown modest efficacy in prior clinical trials, recent data from the SUNLIGHT trial demonstrated that combining FTD–TPI with bevacizumab (FTD–TPI+BEV) may improve overall survival compared to FTD–TPI alone. However, supporting evidence from real-world populations remains scarce. **Methods**: This retrospective study assessed the real-world effectiveness and safety of FTD–TPI+BEV versus FTD–TPI monotherapy in patients with refractory mCRC treated at two institutions from June 2020 to October 2024. **Results**: A total of 106 patients were included, with 47 treated with FTD–TPI+BEV and 59 with FTD–TPI alone. Median progression-free survival (PFS) was significantly longer with FTD–TPI+BEV compared to FTD–TPI alone (4.1 vs. 2.1 months; HR = 0.56; *p* = 0.004), while median overall survival showed a non-significant trend favoring FTD–TPI+BEV (8.4 vs. 6.3 months; HR = 0.74; *p* = 0.189). The disease control rate was also significantly higher with FTD–TPI+BEV (59.6% vs. 25.4%, *p* = 0.001). Subgroup analyses showed consistent PFS benefits. Grade 3–5 adverse events occurred at comparable rates between groups. **Conclusions**: FTD–TPI+BEV may represent a preferred salvage treatment option for refractory mCRC.

## 1. Introduction

Colorectal cancer (CRC) is the fourth most common cancer and the second leading cause of cancer-related mortality worldwide [1]. Approximately 50% of patients with colorectal cancer develop distant metastases over the course of the disease, with a 5-year overall survival rate of only 15% [2]. Standard systemic treatment for metastatic CRC (mCRC) typically consists of fluorouracil-based chemotherapy with oxaliplatin or irinotecan, combined with vascular endothelial growth factor (VEGF)-targeted treatments, such as bevacizumab or epidermal growth factor receptor (EGFR)-targeted therapies, which are used only in *RAS* wild-type tumors [2,3]. Recently, new targeted therapies have significantly improved outcomes in patients with mCRC harboring specific molecular alterations [4]. However, for patients with mCRC who fail standard cytotoxic combination therapy and lack actionable molecular alterations, effective treatment options remain limited. This highlights the urgent need to develop novel therapeutic strategies specifically for this subset of patients.

In the third-line or later treatment setting, trifluridine–tipiracil (FTD–TPI, also known as TAS-102) and regorafenib have demonstrated incremental improvements in median overall survival (OS) compared to placebo and are prescribed as salvage treatment options, as supported by the RECOURSE and CORRECT trials conducted in patients with mCRC refractory to standard chemotherapy [5,6]. More recently, the SUNLIGHT trial showed that FTD–TPI combined with bevacizumab (FTD–TPI+BEV) significantly prolonged survival compared to FTD–TPI alone, establishing it as the current preferred regimen for the treatment of refractory mCRC [7]. However, as the majority of patients in this study (93%) had received treatment only up to the second line and approximately 28% had not previously received anti-VEGF therapy, the applicability of these findings to real-world populations requires careful consideration due to potential differences in clinical characteristics.

A retrospective large-scale study using a claims database examined the effectiveness of FTD–TPI+BEV in real-world settings and demonstrated better OS outcomes compared to FTD–TPI alone or regorafenib [8]. Nevertheless, because the study relied on claims data, specific efficacy outcomes, such as progression-free survival (PFS) and response rate, could not be assessed. In addition, several meta-analyses have consistently suggested that FTD–TPI+BEV is associated with improved survival compared to FTD–TPI monotherapy in patients with refractory mCRC [9,10]. However, because these meta-analyses were based on trials with heterogeneous treatment arms and lacked clinicopathological data beyond survival outcomes, their applicability to real-world patient populations remains limited, making it difficult to identify which subgroups may benefit from the addition of bevacizumab to FTD–TPI.

For most patients with refractory mCRC without druggable molecular alterations, salvage treatments provide only modest effectiveness, and no standard regimen has been established. Given the differences in clinical characteristics between trial populations and real-world patients, along with various clinical factors influencing survival outcomes, evaluating the efficacy of combination therapies in real-world settings is essential. This study aims to assess the real-world effectiveness and safety of FTD–TPI+BEV compared to FTD–TPI alone as a third- or later-line treatment for refractory mCRC.

## 2. Materials and Methods

### 2.1. Patients

Patients with histologically confirmed mCRC were eligible for inclusion if they were refractory or intolerant to fluoropyrimidines, irinotecan, oxaliplatin, and an anti-EGFR monoclonal antibody (for *RAS* wild-type only). Prior exposure to bevacizumab, aflibercept, or regorafenib was permitted. This retrospective study reviewed the medical records of patients with mCRC who experienced treatment failure with standard cytotoxic chemotherapy at Seoul St. Mary’s Hospital and St. Vincent’s Hospital. The study was conducted in accordance with Korean regulatory requirements and the ethical principles outlined in the Declaration of Helsinki. The Institutional Review Board (IRB) of The Catholic University of Korea, Seoul St. Mary’s Hospital, approved the study protocol (approval ID: KC25RISI0178) and granted a waiver of informed consent owing to its retrospective design.

### 2.2. Procedures

Patients received either FTD–TPI+BEV or FTD–TPI alone. FTD–TPI was administered orally at a dose of 35 mg/m^2^ twice daily on days 1–5 and 8–12 of a 28-day cycle. Bevacizumab (5 mg/kg) was administered intravenously on days 1 and 15 of each 28-day cycle. Treatment was continued until radiological or clinical disease progression, unacceptable toxicities, or patient withdrawal. Treatment dose and schedule modifications were allowed to manage adverse events. Dose reductions for FTD–TPI were implemented in a stepwise manner according to the predefined protocol.

### 2.3. Assessments

Tumor evaluations were performed following the Response Evaluation Criteria in Solid Tumors (RECIST) version 1.1. Imaging data for these assessments were obtained through computed tomography or magnetic resonance imaging of the thorax, abdomen, and pelvis conducted at 8-week intervals from the initiation of chemotherapy. Additional imaging was carried out when clinically indicated. As part of the tumor’s evaluation, carcinoembryonic antigen (CEA) levels were measured at baseline and every 8 weeks until disease progression. Adverse events were assessed at each clinic visit and graded based on the National Cancer Institute Common Terminology Criteria for Adverse Events, version 5.0.

### 2.4. Statistical Analysis

Descriptive statistics were presented as proportions for categorical variables and as medians with interquartile ranges (IQRs) for continuous variables. Comparisons of categorical variables were conducted using either the chi-square test or Fisher’s exact test, while continuous variables were analyzed using Student’s *t*-test. OS was defined as the duration from chemotherapy initiation to death from any cause, and PFS was defined as the time from chemotherapy initiation to disease progression or death, whichever occurred first. Patients who did not experience disease progression or death by the data cutoff date were censored, with follow-up duration calculated from chemotherapy initiation to the date of last clinical follow-up or data cutoff. Kaplan–Meier survival curves were generated for each treatment group to estimate median OS and PFS, and differences between groups were assessed using the unstratified log-rank test. Unstratified Cox proportional hazards regression was applied to determine hazard ratios (HRs) and their 95% confidence intervals (CIs). Additionally, a Cox regression model with forward stepwise selection was used to assess the impact of treatment and baseline prognostic factors on survival outcomes. The objective response rate was defined as the proportion of patients who achieved a best overall response of either complete or partial response. The disease control rate was defined as the proportion of patients achieving complete response, partial response, or stable disease, with stable disease required to persist for at least 6 weeks according to RECIST version 1.1 criteria. Fisher’s exact test was used for pairwise comparisons of the objective response rate and the disease control rate between treatment groups. All statistical analyses were two-sided, with *p*-values < 0.05 considered statistically significant. Statistical analysis was performed using SPSS for Windows, version 24.0 (IBM SPSS Inc., Armonk, NY, USA), and GraphPad Prism, version 10.4 (GraphPad Software Inc., San Diego, CA, USA).

## 3. Results

### 3.1. Patients

Between June 2020 and October 2024, a total of 106 patients were included in the study, with 47 treated with FTD–TPI+BEV and 59 treated with FTD–TPI alone. Baseline characteristics of the study population are summarized in Table 1. The median age was 57 years (IQR, 50–64), with no significant difference between the two groups. The proportion of females was higher in the FTD–TPI+BEV group than in the FTD–TPI group (66.0% vs. 40.7%, *p* = 0.010), and patients in the FTD–TPI+BEV group were more likely to have a better Eastern Cooperative Oncology Group (ECOG) performance status (ECOG 0–1: 83.0% vs. 59.3%, *p* = 0.008). A slightly higher proportion of patients in the FTD–TPI+BEV group had a metastatic disease duration of ≤18 months compared to those in the FTD–TPI group (34.0% vs. 18.6%, *p* = 0.071). A higher percentage of patients in the FTD–TPI+BEV group had received only two prior lines of therapy (78.7% vs. 59.3%, *p* = 0.034). Nevertheless, prior exposure to fluoropyrimidine, irinotecan, oxaliplatin, and anti-VEGF therapy, as well as anti-EGFR therapy (for *RAS* wild-type tumors only), was comparable between the two groups.

### 3.2. Effectiveness

The median follow-up duration was 7.3 months (95% CI, 5.6–8.4) based on the reverse Kaplan–Meier method. Among the 106 patients, disease progression was observed in 96 (90.6%), with the remaining 10 patients (9.4%) censored at the time of analysis. For overall survival, 85 patients (80.2%) had died, and the remaining 21 patients (19.8%) were censored. The median PFS for the entire cohort was 2.3 months (95% CI, 1.9–2.7), and the median OS was 7.3 months (95% CI, 5.9–8.7) (Appendix A). By treatment group, the median PFS was 4.1 months (95% CI, 2.8–5.3) in the FTD–TPI+BEV group and 2.1 months (95% CI, 1.9–2.3) in the FTD–TPI group (HR = 0.56, 95% CI, 0.37–0.83; *p* = 0.004, Figure 1A), demonstrating a significantly longer PFS with FTD–TPI+BEV. The median OS was 8.4 months (95% CI, 6.6–10.3) in the FTD–TPI+BEV group and 6.3 months (95% CI, 3.7–8.9) in the FTD–TPI group (HR = 0.74, 95% CI, 0.48–1.15; *p* = 0.189, Figure 1B). Although the FTD–TPI+BEV group exhibited a numerically longer OS, the difference was not statistically significant. Effectiveness outcomes according to treatment regimen are summarized in Table 2. The objective response rates were 2.1% in the FTD–TPI+BEV group and 1.7% in the FTD–TPI group, with no statistically significant difference between the two groups (*p* = 1.000), while the disease control rate was significantly higher in the FTD–TPI+BEV group than in the FTD–TPI group (59.6% vs. 25.4%, *p* = 0.001).

### 3.3. Univariable and Multivariable Analysis for Survival Outcomes

Table 3 presents the findings from the univariable and multivariable analyses conducted to identify predictors of survival outcomes in the entire cohort, including patients treated with either FTD–TPI+BEV or FTD–TPI alone. In the multivariable analysis, peritoneal metastases (HR = 1.66, 95% CI, 1.07–2.58; *p* = 0.023) and a higher baseline neutrophil-to-lymphocyte ratio (NLR) (HR = 1.60, 95% CI, 1.03–2.48; *p* = 0.038) were significantly associated with shorter PFS. Additionally, poor performance status (HR = 1.56, 95% CI, 0.96–2.53; *p* = 0.071) showed a trend toward an association with shorter PFS. Treatment with FTD–TPI+BEV was significantly associated with longer PFS compared to FTD–TPI alone (HR = 0.60, 95% CI, 0.39–0.94; *p* = 0.025). Regarding OS, poor performance status (HR = 3.12, 95% CI, 1.88–5.19; *p* < 0.001), higher baseline NLR (HR = 1.64, 95% CI, 1.04–2.58; *p* = 0.032), and higher baseline CEA (HR = 1.93, 95% CI, 1.18–3.18; *p* = 0.009) were significantly associated with worse OS outcomes.

### 3.4. Subgroup Analysis for Survival Outcomes

Subgroup analyses of PFS and OS comparing the efficacy of FTD–TPI+BEV and FTD–TPI are presented in Figure 2 and Figure 3. The clinical benefits of FTD–TPI+BEV were observed across most subgroups, including those with poor prognostic factors. Patients with *RAS*-mutant disease (HR = 0.45, 95% CI, 0.27–0.74; *p* = 0.001) exhibited a more favorable response to FTD–TPI+BEV compared to those with *RAS* wild-type disease (HR = 0.85, 95% CI, 0.43–1.70; *p* = 0.647). Similarly, the treatment effect was more evident in patients with ≥3 organ metastatic sites (HR = 0.36, 95% CI, 0.21–0.63; *p* < 0.001), whereas those with ≤2 metastatic sites showed minimal benefit (HR = 0.91, 95% CI, 0.49–1.67; *p* = 0.747). Additionally, the addition of bevacizumab to FTD–TPI demonstrated clinical benefits regardless of whether the bevacizumab-free interval was shorter or longer than six months. For OS, although the differences were not statistically significant, FTD–TPI+BEV showed a trend toward improved outcomes compared to FTD–TPI alone in most subgroups. However, in patients with poor performance status, FTD–TPI+BEV was associated with worse OS outcomes (HR = 1.75, 95% CI, 0.68–4.49; *p* = 0.157). Furthermore, OS outcomes were comparable between the two treatment groups in patients with ≤2 organ metastatic sites (HR = 0.89, 95% CI, 0.44–1.80; *p* = 0.736) and those with a bevacizumab-free interval of ≤6 months (HR = 0.93, 95% CI, 0.52–1.65; *p* = 0.791).

### 3.5. Safety

Grade 3–5 adverse events occurred in 25 patients (53.2%) in the FTD–TPI+BEV group and 32 patients (54.2%) in the FTD–TPI group (Table 4). The most frequently reported grade 3–5 adverse event was neutropenia, which affected 21 patients (38.6%) in the FTD–TPI+BEV group and 23 patients (39.0%) in the FTD–TPI group, with no notable difference between treatment groups. Febrile neutropenia was reported in two patients (4.2%) in the FTD–TPI+BEV group and two patients (3.4%) in the FTD–TPI group. Grade 3 or higher anemia was more frequently observed in the FTD–TPI group (18.6%) than in the FTD–TPI+BEV group (8.4%). Grade 1–2 nausea and fatigue occurred more frequently in the FTD–TPI+BEV group, with nausea reported in 14 patients (29.8%) compared to 8 patients (13.6%) in the FTD–TPI group and fatigue in 11 patients (23.4%) compared to 10 patients (16.9%).

## 4. Discussion

This retrospective study compared the overall effectiveness and safety of FTD–TPI+BEV and FTD–TPI monotherapy in a real-world setting for patients with mCRC refractory to standard chemotherapy. FTD–TPI+BEV demonstrated a significantly longer PFS compared to FTD–TPI monotherapy, along with a significantly higher disease control rate. However, although the FTD–TPI+BEV group exhibited a trend toward longer OS than the FTD–TPI monotherapy group, the difference was not statistically significant. Regardless of whether patients were treated with FTD–TPI+BEV or FTD–TPI alone, poor performance status, higher baseline NLR, and elevated CEA levels were significantly associated with worse OS outcomes. Additionally, FTD–TPI+BEV showed improved effectiveness outcomes across most subgroups compared to FTD–TPI monotherapy, with no significantly higher incidence of adverse events observed in the combination therapy group.

The overall study population had a similar proportion of *RAS*-mutated patients (60–70%) compared to previous trials evaluating FTD–TPI+BEV and FTD–TPI monotherapy, including the SUNLIGHT trial and a phase II trial [7,11]. However, as this was a real-world study, it included a higher proportion of patients with poor performance status (ECOG 2, 30.2%) and more heavily pretreated patients, with 40.7% having received ≥3 prior lines of therapy for metastatic disease. Furthermore, almost all patients (97.2%) had previously received anti-VEGF therapy, a markedly higher proportion than in prior trials, representing a key distinction from those studies.

The addition of bevacizumab to FTD–TPI provided a clinically meaningful advantage over FTD–TPI monotherapy in disease control and survival outcomes. However, the difference in OS was not statistically significant, which may be attributed to the lower incidence of death events in the FTD–TPI+BEV group, resulting in a substantial proportion of patients being censored (event of death: FTD–TPI+BEV vs. FTD–TPI, 54.8% vs. 100%). Given these findings, extended follow-up may provide further maturation of OS data, potentially revealing a statistically significant difference between the two groups. In patients with mCRC refractory to standard chemotherapy, prognostic factors, such as CEA, which reflects tumor burden and performance status, appeared to have a greater influence on survival outcomes. Consistent with findings from the RECOURSE trial [12], our study also identified a high NLR as an independent prognostic factor associated with inferior PFS and OS. Furthermore, the presence of peritoneal metastases was significantly correlated with worse PFS outcomes, suggesting that the effectiveness of oral chemotherapy may be limited in this subgroup of patients.

In the subgroup analysis, FTD–TPI+BEV demonstrated superior clinical efficacy compared to FTD–TPI monotherapy, regardless of primary tumor location, *RAS* mutation status, duration of metastatic disease, or number of prior lines of therapy. Notably, the survival benefit of the combination therapy was observed irrespective of the length of the bevacizumab-free interval. Consistent with prior evidence demonstrating that the continuation of bevacizumab in second-line chemotherapy improves survival outcomes after progression on first-line doublet plus bevacizumab regimens [13], our findings suggest that FTD–TPI+BEV may still offer clinical benefits even in patients who were refractory to a bevacizumab-containing regimen in their most recent treatment. However, the advantage of FTD–TPI+BEV appeared to be less pronounced in patients with poor performance status or lower tumor burden, such as those with fewer metastatic organ sites.

From a safety perspective, our findings differed from those of the SUNLIGHT trial. The addition of bevacizumab to FTD–TPI was not associated with a higher incidence of severe neutropenia, and no other clinically meaningful differences in adverse events were observed between the treatment groups. Notably, grade 3 or higher anemia was more frequently observed in the FTD–TPI monotherapy group, which may be attributable to the inclusion of patients for whom bevacizumab was contraindicated, such as those with comorbid conditions like gastrointestinal bleeding. Non-hematologic toxicities of any grade, including nausea and fatigue, were more commonly reported in the FTD–TPI+BEV group, which may be related to the longer median treatment duration in this group compared to the FTD–TPI group.

For patients with mCRC refractory to chemotherapy and without biomarkers for targeted therapy, salvage treatment options may include FTD–TPI with or without bevacizumab, regorafenib, or fruquintinib. While regorafenib, FTD–TPI, and fruquintinib have demonstrated OS benefit over placebo in previous clinical trials, FTD–TPI+BEV has shown superior OS compared to an active control, FTD–TPI monotherapy [5,6,7,14]. Given these findings, FTD–TPI in combination with bevacizumab may be regarded as the preferred salvage regimen in patients with good performance status. In our study, FTD–TPI+BEV demonstrated improved effectiveness outcomes regardless of the number of prior treatment lines or previous exposure to bevacizumab, supporting its use as a viable option in real-world clinical settings.

In patients for whom bevacizumab is contraindicated, such as those with a history of severe hemorrhage, impaired wound healing, or fistula formation, alternative options, including FTD–TPI monotherapy, regorafenib, or fruquintinib may be considered. Treatment selection should be made in careful consideration of the patient’s clinical condition and the toxicity profile of each agent. FTD–TPI monotherapy, as observed in our real-world study, was associated with considerable hematological toxicities and should be used with caution in patients with a history of recurrent systemic infections. In contrast to FTD–TPI, regorafenib has limited hematologic toxicity but is associated with other adverse events, such as hand–foot skin reaction, proteinuria, and hypertension [6]. Nearly half of patients experience hand–foot skin reactions, underscoring the need to evaluate individual tolerability before initiating treatment [15]. Fruquintinib demonstrated a survival benefit and manageable toxicity in the FRESCO-2 trial in patients with mCRC who had progressed on or were intolerant to FTD–TPI or regorafenib [14]. Given its relatively favorable safety profile, fruquintinib may be a suitable alternative for patients with a lower disease burden and an indolent disease course who are unable to tolerate FTD–TPI or regorafenib.

Our study has several limitations. First, because the combination of bevacizumab with FTD–TPI was introduced as a treatment option following the SUNLIGHT trial results, the follow-up duration for patients treated with FTD–TPI plus bevacizumab was relatively short. Consequently, fewer death events occurred in this group, potentially limiting our ability to detect statistically significant differences in overall survival between treatment arms. Second, the relatively small patient cohort and the multiple statistical comparisons conducted increased the risk of type I errors; thus, statistically significant findings, especially those approaching significance thresholds, require careful interpretation. Moreover, the limited sample size reduced the statistical power of subgroup analyses, resulting in wide confidence intervals and decreased reliability. Additionally, it precluded advanced statistical methods, such as propensity score matching. Third, some patients experienced rapid clinical deterioration, precluding comprehensive radiologic assessment of treatment response and potentially confounding the evaluation of treatment effectiveness. Future studies with larger patient cohorts allowing for thorough radiologic assessments and employing rigorous statistical methods will be necessary to validate our findings and clarify treatment benefits.

In conclusion, our study demonstrated the effectiveness and safety of FTD–TPI with or without bevacizumab in a real-world setting among patients with chemorefractory mCRC. FTD–TPI+BEV was associated with improved survival outcomes compared to FTD–TPI monotherapy across most subgroups. Considering the varying prognostic factors, FTD–TPI+BEV may be a suitable option, particularly for selected patients with good performance status who are medically fit. Further prospective studies incorporating molecular biomarker analyses are required to better define subgroups most likely to benefit from FTD–TPI+BEV and to refine treatment sequencing strategies, including other oral chemotherapies, within the continuum of care.

## Figures and Tables

**Figure 1 biomedicines-13-00976-f001:**
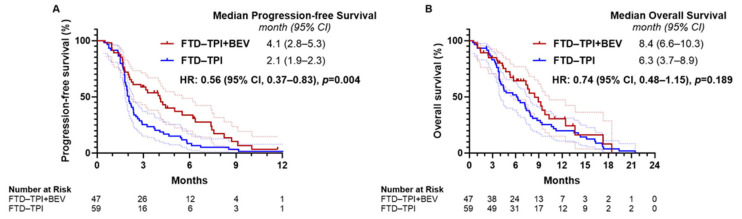
Kaplan–Meier estimates of (**A**) progression-free survival and (**B**) overall survival by treatment group.

**Figure 2 biomedicines-13-00976-f002:**
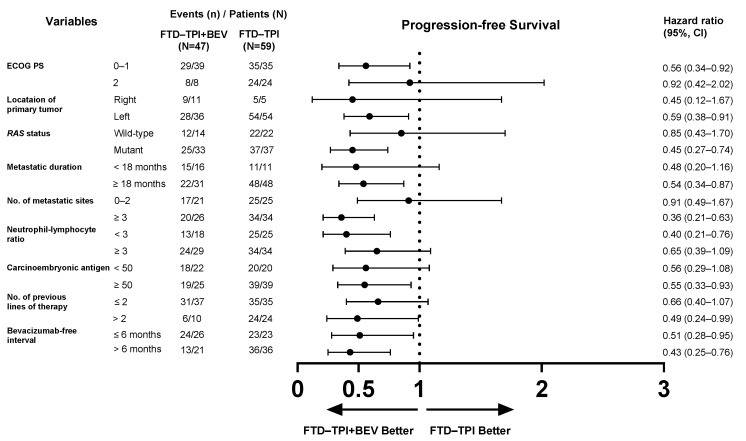
Forest plot of subgroup analyses for progression-free survival.

**Figure 3 biomedicines-13-00976-f003:**
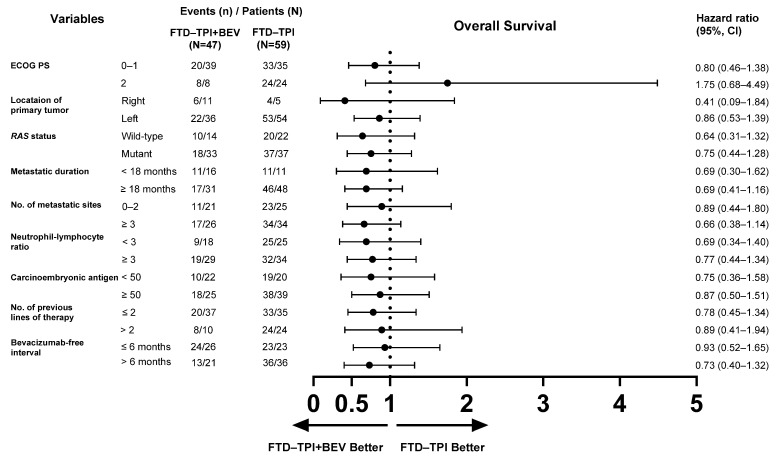
Forest plot of subgroup analyses for overall survival.

**Table 1 biomedicines-13-00976-t001:** Baseline characteristics.

Variable	Total (*n* = 106)	FTD–TPI+BEV (*n* = 47)	FTD–TPI (*n* = 59)	*p* Value
**Age, years**	57 (50–64)	55 (49–62)	59 (50–64)	0.324
≥65 yr	22 (20.8)	9 (19.1)	13 (22.0)	0.716
**Gender**				
Male	51 (48.1)	16 (34.0)	35 (59.3)	0.010
Female	55 (51.9)	31 (66.0)	24 (40.7)	
**ECOG performance status**				
0–1	74 (69.8)	39 (83.0)	35 (59.3)	0.008
2	32 (30.2)	8 (17.0)	24 (40.7)	
**Primary diagnosis**				
Colon cancer	65 (61.3)	31 (66.0)	34 (57.6)	0.382
Rectal cancer	41 (38.7)	16 (34.0)	25 (42.4)	
**Primary tumor location**				
Right side	16 (15.1)	11 (23.4)	5 (8.5)	0.054
Left side	90 (84.9)	36 (76.6)	54 (91.5)	
**Histology**				
Adenocarcinoma	99 (93.4)	46 (97.9)	53 (89.8)	0.129
Mucinous carcinoma	7 (6.6)	1 (2.1)	6 (10.2)	
**Duration of metastatic disease**				
Median, months	27.2 (17.9–49.3)	23.9 (17.3–51.9)	29.1 (19.2–48.4)	0.876
<18 months	27 (25.5)	16 (34.0)	11 (18.6)	0.071
≥18 months	79 (74.5)	31 (66.0)	48 (81.4)	
**Number of metastatic organ sites**				
1 or 2	46 (43.4)	21 (44.7)	25 (42.4)	0.812
≥3	60 (56.6)	26 (55.3)	34 (57.6)	
** *RAS* ** ** mutation status**				
Wild-type	36 (34.0)	14 (29.8)	22 (37.3)	0.418
Mutant	70 (66.0)	33 (70.2)	37 (62.7)	
** *BRAF* ** ** mutation status**				
Wild-type	106 (100)	47 (100)	59 (100)	
**MMR and MSI status**				
MMR proficient or MSI stable	106 (100)	47 (100)	59 (100)	
**Previous lines of therapy ***				
2	72 (67.9)	37 (78.7)	35 (59.3)	0.034
≥3	34 (40.7)	10 (21.3)	24 (40.7)	
**Previous treatments**				
Fluoropyrimidine	106 (100)	47 (100)	59 (100)	
Irinotecan	106 (100)	47 (100)	59 (100)	
Oxaliplatin	106 (100)	47 (100)	59 (100)	
Anti-VEGF therapy	103 (97.2)	45 (95.7)	58 (98.3)	
Anti-EGFR therapy ^†^	35 (97.2)	14 (100)	21 (95.4)	
**Neutrophil to lymphocyte ratio**				
<3	43 (40.6)	18 (38.3)	25 (42.4)	0.671
≥3	63 (59.4)	29 (61.7)	34 (57.6)	
**Baseline CEA**				
<50 μg/L	42 (39.6)	22 (46.8)	20 (33.9)	0.177
≥50 μg/L	64 (60.4)	25 (53.2)	39 (66.1)	

FTD–TPI+BEV, trifluridine–tipiracil plus bevacizumab; FTD–TPI, trifluridine–tipiracil; ECOG, Eastern Cooperative Oncology Group; MMR, mismatch repair; MSI, microsatellite instability; VEGF, vascular endothelial growth factor; EGFR, epidermal growth factor receptor; CEA, carcinoembryonic antigen. Data are *n* (%) or median (IQR). * Systemic treatment for metastatic disease, including cytotoxic chemotherapy, targeted therapy, and receptor tyrosine kinase inhibitors. ^†^ Proportion of patients with *RAS* wild-type disease.

**Table 2 biomedicines-13-00976-t002:** Effectiveness outcomes of trifluridine–tipiracil with bevacizumab compared to trifluridine–tipiracil alone in patients with metastatic colorectal cancer.

Variable	FTD–TPI+BEV (*n* = 47)	FTD–TPI (*n* = 59)	*p* Value
Best overall response, *n* (%)			
Partial response	1 (2.1)	1 (1.7)	
Stable disease	27 (57.4)	14 (23.7)	
Progressive disease	19 (40.4)	44 (74.6)	
Objective response rate, *n* (%)	1 (2.1)	1 (1.7)	1.000
Disease control rate, *n* (%)	28 (59.6)	15 (25.4)	0.001
Median PFS, months [95% CI]	4.1 [2.8–5.3]	2.1 [1.9–2.3]	
16-week PFS, % [95% CI]	51.0 [35.5–64.6]	20.3 [11.2–31.3]	
Median OS, months [95% CI]	8.4 [6.6–10.3]	6.3 [3.7–8.9]	
9-month OS, % [95% CI]	45.7 [28.1–61.7]	29.0 [17.9–41.0]	

FTD–TPI+BEV, trifluridine–tipiracil plus bevacizumab; FTD–TPI, trifluridine–tipiracil; PFS, progression-free survival; OS, overall survival.

**Table 3 biomedicines-13-00976-t003:** Univariable and multivariable analyses of clinicopathologic features associated with progression-free survival and overall survival in patients with metastatic colorectal cancer.

	PFS				OS			
Variables	Univariable Analysis		Multivariable Analysis		Univariable Analysis		Multivariable Analysis	
	HR (95% CI)	*p* Value	HR (95% CI)	*p* Value	HR (95% CI)	*p* Value	HR (95% CI)	*p* Value
Age ≥ 65 yr (vs. <65 yr)	0.84 (0.51–1.38)	0.498			1.11 (0.61–2.01)	0.734		
Female (vs. male)	0.84 (0.56–1.27)	0.413			1.17 (0.76–1.80)	0.483		
ECOG PS 2 (vs. PS 0–1)	2.11 (1.36–3.27)	0.001	1.56 (0.96–2.53)	0.071	3.30 (2.06–5.29)	<0.001	3.12 (1.88–5.19)	<0.001
Rectal (vs. colon)	1.31 (0.87–1.98)	0.198			1.26 (0.81–1.96)	0.306		
Right side (vs. left side)	0.78 (0.44–1.38)	0.393			0.70 (0.36–1.36)	0.294		
*RAS* mutant (vs. wild-type)	1.02 (0.67–1.56)	0.934			1.13 (0.72–1.77)	0.600		
Previous lines of therapy >2 (vs. ≤2)	1.31 (0.84–2.06)	0.231			1.38 (0.88–2.15)	0.158		
Metastatic duration (<18 vs. ≥18 months)	1.05 (0.67–1.66)	0.833			0.92 (0.56–1.51)	0.735		
No. of sites of metastasis (≥3 vs. 0–2)	1.25 (0.83–1.88)	0.281			1.50 (0.96–2.34)	0.072		
Liver metastases (vs. none)	1.44 (0.93–2.24)	0.105			0.96 (0.61–1.50)	0.844		
Lung metastases (vs. none)	0.90 (0.57–1.40)	0.629			1.22 (0.75–1.99)	0.415		
Peritoneum metastases (vs. none)	1.71 (1.12–2.61)	0.013	1.66 (1.07–2.58)	0.023	1.24 (0.80–1.92)	0.342		
Neutrophil–lymphocyte ratio ≥3 (vs. <3)	1.63 (1.08–2.48)	0.022	1.60 (1.03–2.48)	0.038	1.79 (1.15–2.78)	0.010	1.64 (1.04–2.58)	0.032
CEA ≥50 μg/L (vs. <50 μg/L)	1.56 (1.03–2.37)	0.038	1.21 (0.77–1.91)	0.406	2.41 (1.50–3.87)	<0.001	1.93 (1.18–3.18)	0.009
FTD–TPI+BEV (vs. FTD–TPI)	0.56 (0.37–0.83)	0.004	0.60 (0.39–0.94)	0.025	0.74 (0.48–1.15)	0.189	1.09 (0.66–1.78)	0.742

PFS, progression-free survival; OS, overall survival; HR, hazard ratio; ECOG PS, Eastern Cooperative Oncology Group performance status; CEA, carcinoembryonic antigen; FTD–TPI+BEV, trifluridine–tipiracil plus bevacizumab; FTD–TPI, trifluridine–tipiracil.

**Table 4 biomedicines-13-00976-t004:** Adverse events.

	FTD–TPI+BEV (*n* = 47)		FTD–TPI (*n* = 59)	
	Any Grade	Grade ≥ 3	Any Grade	Grade ≥ 3
Any	43 (91.5)	25 (53.2)	49 (83.0)	32 (54.2)
Diarrhea	0	0	5 (8.5)	2 (3.4)
Nausea	14 (29.8)	2 (4.2)	8 (13.6)	1 (1.7)
Vomiting	6 (12.8)	1 (2.1)	7 (11.9)	1 (1.7)
Fatigue	11 (23.4)	1 (2.1)	10 (16.9)	1 (1.7)
Neutropenia	28 (59.6)	21 (38.6)	30 (50.8)	23 (39.0)
Febrile neutropenia	2 (4.2)	2 (4.2)	2 (3.4)	2 (3.4)
Anemia	25 (53.2)	4 (8.4)	34 (57.6)	11 (18.6)
Thrombocytopenia	9 (19.1)	2 (4.2)	14 (23.7)	4 (6.8)

FTD–TPI+BEV, trifluridine–tipiracil plus bevacizumab; FTD–TPI, trifluridine–tipiracil. Data are number of patients (%).

## Data Availability

All materials (data and images) reported in this article are available within the paper and its Appendix A.

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
