# Peer review of "Real-World Comparison of Trifluridine–Tipiracil with or Without Bevacizumab in Patients with Refractory Metastatic Colorectal Cancer"

_biomedicines, 2025, doi:10.3390/biomedicines13040976_

Round 1

Reviewer 1 Report

Comments and Suggestions for Authors

Biomedicines, Hyunho Kim et al., 2025, methodological review

The data presented in this study confirm by and large the  results of the randomized SUNLIGHT trial and other previously published data series on the FTD-PPI + bevacizumab combination. However, from a biostatistical point-of-view, the manuscript suffers from major limitations.

Major issues:

While the statistical techniques described are generally appropriate for this type of retrospective comparative studies, the most important limitation of the study is the low patient number, leading to a high level of statistical uncertainty (type I and II error), not sufficiently reflected in the text.

If the number of p values (around 70 in this manuscript) is in the range of the sample size (n=106), the risk of type I errors (false positive findings) is highly inflated. Thus, the “standard” statement “All statistical analyses were two-sided, with p-values < .05 considered statistically significant” (l. 125) is not acceptable without addressing this issue as major limitation of the study.

Moreover, the limited patient/event numbers lead to insufficient power, namely with respect to the OS results. The large HR confidence interval of 0.74 (0.48-1.15) widely overlaps the HR estimate of SUNLIGHT (0.61), and vice versa. As the effect of adding the VEGF antibody is the main research question of this study, this variable should explicitly remain included in the multivariable Cox model, independent of its univariable p value (Table 3, right side).

In light of the low power, the extensive reliance on subgroup analyses is problematic and questionable, as can be easily derived from the large confidence intervals of all estimates in both Fig.2. Conclusions drawn from these data are generally not warranted. E.g., the subgroup result for patients with poor ECOG status (even reflected in the concluding sentence of the abstract!) is based on 8(!) patients only, with a confidence interval for OS and PFS easily including the finding of the complementary group with ECOG 0-1.

Minor issues:

  • 29 and others: The “disease-control rate” is not clearly defined (required duration of SD?).
  • The calculation method for the median follow-up duration of 5.9 mo should be provided.
  • As the issue of limited OS follow-up is mentioned at several points in the manuscript, it would be helpful to add the patients-at-risk numbers below the x axes in Fig. 1.
  • Throughout the text “multivariate” should be replaced by the more exact term “multivariable”.
  • “Figure 2” is used twice.

In conclusion, due to the low patient and event numbers, these „real-world“ data provide only a small gain of information in addition to the evidence from the SUNLIGHT trial or the data from Kagawa et al., except for the aspect that quite similar treatment effects can been detected in VEGF-pretreated patients. Any conclusions based on the subgroup analyses are highly speculative.

Author Response

Reviewer 1

The data presented in this study confirm by and large the results of the randomized SUNLIGHT trial and other previously published data series on the FTD-PPI + bevacizumab combination. However, from a biostatistical point-of-view, the manuscript suffers from major limitations.

Comment 1: While the statistical techniques described are generally appropriate for this type of retrospective comparative studies, the most important limitation of the study is the low patient number, leading to a high level of statistical uncertainty (type I and II error), not sufficiently reflected in the text. If the number of p values (around 70 in this manuscript) is in the range of the sample size (n=106), the risk of type I errors (false positive findings) is highly inflated. Thus, the “standard” statement “All statistical analyses were two-sided, with p-values < .05 considered statistically significant” (l. 125) is not acceptable without addressing this issue as major limitation of the study.

Response 1: Thank you for your insightful and valuable comments. We fully acknowledge your concerns regarding the elevated risk of type I errors due to our limited sample size (n=106) and the relatively large number of statistical comparisons (approximately 70 p-values reported). In real-world clinical practice, however, the number of patients treated with FTD–TPI, with or without bevacizumab, is relatively small, and comprehensive clinicopathological data are often limited. Thus, we inevitably conducted our analysis with a small sample size, which underscores the need for cautious interpretation of our findings. To clearly reflect this methodological concern, we have explicitly addressed the increased potential for type I error in the revised manuscript’s limitations section, highlighting the need for cautious interpretation of statistically significant findings, particularly those near the threshold of significance. Furthermore, we strongly agree that larger-scale, multicenter observational studies with greater statistical power and more comprehensive clinical data are warranted to validate and extend our results. Thank you again for highlighting this important methodological consideration.

The manuscript is revised as follows:

Our study has several limitations. First, because the combination of bevacizumab with FTD–TPI was introduced as a treatment option following the SUNLIGHT trial results, the follow-up duration for patients treated with FTD–TPI plus bevacizumab was relatively short. Consequently, fewer death events occurred in this group, potentially limiting our ability to detect statistically significant differences in overall survival between treatment arms. Second, the relatively small patient cohort and the multiple statistical comparisons conducted increased the risk of type I errors; thus, statistically significant findings, especially those approaching significance thresholds, require careful interpretation. Moreover, the limited sample size reduced statistical power for subgroup analyses, resulting in wide confidence intervals and decreased reliability. Additionally, it precluded advanced statistical methods, such as propensity-score matching. Third, some patients experienced rapid clinical deterioration, precluding comprehensive radiologic assessment of treatment response and potentially confounding the evaluation of treatment effectiveness. Future studies with larger patient cohorts, allowing for thorough radiologic assessments and employing rigorous statistical methods, will be necessary to validate our findings and clarify treatment benefit.

Comment 2: Moreover, the limited patient/event numbers lead to insufficient power, namely with respect to the OS results. The large HR confidence interval of 0.74 (0.48-1.15) widely overlaps the HR estimate of SUNLIGHT (0.61), and vice versa. As the effect of adding the VEGF antibody is the main research question of this study, this variable should explicitly remain included in the multivariable Cox model, independent of its univariable p value (Table 3, right side).

Response 2: Thank you for your insightful comments. We fully agree with your points regarding the limited statistical power due to the relatively small number of patients and events, particularly concerning the overall survival results. We acknowledge that the wide confidence interval of our observed hazard ratio considerably overlaps with the previously reported findings of the SUNLIGHT trial, thus limiting the interpretation of our results. We have explicitly addressed these issues in the limitations section of the revised manuscript.

Furthermore, we completely agree with your recommendation that, since evaluating the benefit of adding bevacizumab to FTD–TPI was one of the primary objectives of our study, this variable should remain in the multivariate Cox regression model irrespective of its statistical significance in univariate analysis. Accordingly, we have performed additional analyses as you suggested and have clearly described these adjustments in our revised manuscript.

The manuscript is revised as follows:

Treatment with FTD–TPI+BEV was significantly associated with longer PFS compared to FTD–TPI alone (HR = 0.60; 95% CI, 0.39–0.94; p = 0.025). Regarding OS, poor performance status (HR = 3.12; 95% CI, 1.88–5.19; p < 0.001), higher baseline NLR (HR = 1.64; 95% CI, 1.04–2.58; p = 0.032), and higher baseline CEA (HR = 1.93; 95% CI, 1.18–3.18; p = 0.009) were significantly associated with worse OS outcomes.

PFS

OS

Variables

Univariate analysis

Multivariate analysis

Univariate analysis

Multivariate analysis

HR (95% CI)

p value

HR (95% CI)

p value

HR (95% CI)

p value

HR (95% CI)

p value

 Age ≥65 yr (vs. <65 yr)

0.84 (0.51–1.38)

0.498

1.11 (0.61–2.01)

0.734

Female (vs. male)

0.84 (0.56–1.27)

0.413

1.17 (0.76–1.80)

0.483

ECOG PS 2 (vs. PS 01)

2.11 (1.36–3.27)

0.001

1.56 (0.96–2.53)

0.071

3.30 (2.06–5.29)

<0.001

3.12 (1.88–5.19)

<0.001

Rectal (vs. Colon)

1.31 (0.87–1.98)

0.198

1.26 (0.81–1.96)

0.306

Right side (vs. Left side)

0.78 (0.44–1.38)

0.393

0.70 (0.36–1.36)

0.294

RAS Mutant (vs. Wild-type)

1.02 (0.67–1.56)

0.934

1.13 (0.72–1.77)

0.600

Previous lines of therapy >2 (vs. ≤2)

1.31 (0.84–2.06)

0.231

1.38 (0.88–2.15)

0.158

Metastatic duration (<18 vs. ≥18 months)

1.05 (0.67–1.66)

0.833

0.92 (0.56–1.51)

0.735

No. of sites of metastasis (≥3 vs. 0-2)

1.25 (0.83–1.88)

0.281

1.50 (0.96–2.34)

0.072

Liver metastases (vs. none)

1.44 (0.93–2.24)

0.105

0.96 (0.61–1.50)

0.844

Lung metastases (vs. none)

0.90 (0.57–1.40)

0.629

1.22 (0.75–1.99)

0.415

Peritoneum metastases (vs. none)

1.71 (1.12–2.61)

0.013

1.66 (1.07–2.58)

0.023

1.24 (0.80–1.92)

0.342

Neutrophil-lymphocyte ratio ≥3 (vs. <3)

1.63 (1.08–2.48)

0.022

1.60 (1.03–2.48)

0.038

1.79 (1.15–2.78)

0.010

1.64 (1.04–2.58)

0.032

CEA ≥50 μg/L (vs. <50 μg/L)

1.56 (1.03–2.37)

0.038

1.21 (0.77–1.91)

0.406

2.41 (1.50–3.87)

<0.001

1.93 (1.18–3.18)

0.009

FTD–TPI+BEV (vs. FTD–TPI)

0.56 (0.37–0.83)

0.004

0.60 (0.39–0.94)

0.025

0.74 (0.48–1.15)

0.189

1.09 (0.66–1.78)

0.742

Comment 3: In light of the low power, the extensive reliance on subgroup analyses is problematic and questionable, as can be easily derived from the large confidence intervals of all estimates in both Fig.2. Conclusions drawn from these data are generally not warranted. E.g., the subgroup result for patients with poor ECOG status (even reflected in the concluding sentence of the abstract!) is based on 8(!) patients only, with a confidence interval for OS and PFS easily including the finding of the complementary group with ECOG 0-1.

Response 3: Thank you for raising this critical methodological issue. We fully acknowledge your concern regarding the extensive subgroup analyses performed given the limited sample size, which resulted in wide confidence intervals and reduced statistical power. We agree that our subgroup analyses should be viewed as exploratory and hypothesis-generating rather than definitive. Therefore, we have explicitly clarified this limitation in the revised manuscript’s discussion and limitation sections, emphasizing cautious interpretation. Additionally, we revised the abstract and conclusions to avoid overinterpreting subgroup results based on very small patient numbers (e.g., poor ECOG subgroup). Future studies with larger sample sizes will be necessary to validate these exploratory findings.

The manuscript is revised as follows:

Disease control rate was also significantly higher with FTD–TPI+BEV (59.6% vs. 25.4%, p = 0.001). Subgroup analyses showed consistent PFS benefits. Grade 3–5 adverse events occurred at com-parable rates between groups. FTD–TPI+BEV may represent a preferred salvage treatment option for refractory mCRC.

Our study has several limitations. First, because the combination of bevacizumab with FTD–TPI was introduced as a treatment option following the SUNLIGHT trial results, the follow-up duration for patients treated with FTD–TPI plus bevacizumab was relatively short. Consequently, fewer death events occurred in this group, potentially limiting our ability to detect statistically significant differences in overall survival between treatment arms. Second, the relatively small patient cohort and the multiple statistical comparisons conducted increased the risk of type I errors; thus, statistically significant findings, especially those approaching significance thresholds, require careful interpretation. Moreover, the limited sample size reduced statistical power for subgroup analyses, resulting in wide confidence intervals and decreased reliability. Additionally, it precluded advanced statistical methods, such as propensity-score matching. Third, some patients experienced rapid clinical deterioration, precluding comprehensive radiologic assessment of treatment response and potentially confounding the evaluation of treatment effectiveness. Future studies with larger patient cohorts, allowing for thorough radiologic assessments and employing rigorous statistical methods, will be necessary to validate our findings and clarify treatment benefit.

In conclusion, Our study demonstrated the effectiveness and safety of FTD–TPI with or without bevacizumab in a real-world setting among patients with chemorefractory mCRC. FTD–TPI+BEV was associated with improved survival outcomes com-pared to FTD–TPI monotherapy across most subgroups. Considering the varying prognostic factors, FTD–TPI+BEV may be a suitable option, particularly for selected patients with good performance status who are medically fit. Further prospective studies, incorporating molecular biomarker analyses, are required to better define subgroups most likely to benefit from FTD–TPI+BEV and refine treatment sequencing strategies, including other oral chemotherapies, within the continuum of care.

Comment 4: 29 and others: The “disease-control rate” is not clearly defined (required duration of SD?).

Response 4: We truly appreciate your thorough review. We fully agree that the definition of disease-control rate needed clarification. Accordingly, we have explicitly defined disease-control rate in the revised manuscript as the proportion of patients achieving complete response, partial response, or stable disease, with stable disease required to persist for at least 6 weeks according to RECIST v1.1 criteria.

The manuscript is revised as follows:

Additionally, a Cox regression model with forward stepwise selection was used to assess the impact of treatment and baseline prognostic factors on survival outcomes. Objective response rate was defined as the proportion of patients achieving a best overall response of either complete or partial response. Disease control rate was defined as the proportion of patients achieving complete response, partial response, or stable disease, with stable disease required to persist for at least 6 weeks according to RECIST version 1.1 criteria.

Comment 5: The calculation method for the median follow-up duration of 5.9 mo should be provided.

Response 5: We truly appreciate your thorough review. Initially, we calculated the median follow-up duration using a simple median of follow-up times. However, given that a substantial proportion of patients were censored for the OS outcome in our study, we believe that the reverse Kaplan-Meier method is a more appropriate approach for estimating the median follow-up duration. Accordingly, we have recalculated the median follow-up using this method and revised the manuscript to reflect this change.

The manuscript is revised as follows:

The median follow-up duration was 7.3 months (95% CI, 5.6–8.4) based on the reverse Kaplan–Meier method.

Comment 6: As the issue of limited OS follow-up is mentioned at several points in the manuscript, it would be helpful to add the patients-at-risk numbers below the x axes in Fig. 1.

Response 6: We sincerely appreciate your thorough review. We fully agree with your suggestion regarding the necessity to include the number at risk in the Kaplan-Meier curves for PFS and OS. We have revised the figures accordingly. Thank you for pointing this out.

The manuscript is revised as follows:

Comment 7: Throughout the text “multivariate” should be replaced by the more exact term “multivariable”.

Response 7: Thank you for highlighting this important distinction. We fully agree with your comment and have replaced the term "multivariate" with the more precise term "multivariable" throughout our manuscript, as the analyses conducted involve a single dependent variable analyzed in relation to multiple independent variables.

The manuscript is revised as follows:

3.3. Univariable and Multivariable Analysis for Survival Outcomes

Table 3. Univariable and multivariable analyses of clinicopathologic features associated with progression-free survival and overall survival in patients with metastatic colorectal cancer.

Comment 8:  “Figure 2” is used twice.

Response 8: Thank you for your thoughtful comment. We have corrected the mislabeled part accordingly.

Comment 8: In conclusion, due to the low patient and event numbers, these „real-world“ data provide only a small gain of information in addition to the evidence from the SUNLIGHT trial or the data from Kagawa et al., except for the aspect that quite similar treatment effects can been detected in VEGF-pretreated patients. Any conclusions based on the subgroup analyses are highly speculative.

Response 8: Thank you very much for highlighting these important considerations. We fully acknowledge your concerns regarding the limited patient and event numbers, which indeed constrain the additional insights our real-world study can provide beyond the robust evidence from the SUNLIGHT trial and prior studies such as Kagawa et al. We agree that the primary incremental value of our data may lie in demonstrating similar treatment efficacy of FTD–TPI plus bevacizumab even in patients previously treated with VEGF inhibitors, a finding less clearly addressed by prior trials.

Furthermore, as you have emphasized, we fully agree that the conclusions drawn from our subgroup analyses are exploratory and speculative due to the limited sample size, and we have accordingly revised the manuscript to reflect a more cautious interpretation.

Reviewer 2 Report

Comments and Suggestions for Authors

This is a retrospective, multicenter (2 centers) study with the aim to investigate the efficacy and safety of two treatments (FTD-TPI+B vs FTD-TPI) as third or later-line treatment for refractory metastatic colorectal cancer patients.

One hundred and six patients were selected between June 2020 and October 2024.

The Authors find that patients treated with FTD-TPI+B had longer median progression-free survival and higher disease control rate compared to patients treated with FTD-TPI alone. Overall survival and safety were similar in the two treatments.

The conclusion is that FTD-TPI+B should be considered for patients with good performance status (0-1).

Comments.

The study is interesting, however there are some points to clarify.

The main criticism is that this is a retrospective study, not randomized, thus the comparison between treatments should not be made.

Moreover, the effectiveness should be considered instead of the efficacy.

A propensity-score matching samples should be performed, matching patients treated with the two treatments for some prognostic factors (see Austin PC. The use of propensity score methods with survival or time-to-event outcomes: reporting measures of effect similar to those used in randomized experiments. Stat Med 2014;33(7):1242-1258. doi: 10.1002/sim.5984)

The NLR with a cut-off of 3 was used, but how this cut-off was chosen?

Please reports the numbers of patients at risk under the X-axis of each figure.

In the statistical analysis, PFS and OS were defined but patients censored at the time of analysis were not reported. How the time of patients not progressed/death was considered?

Please report univariate and multivariate analyses according to age in Table 3.

In Table 3, the site of metastases was reported (liver, lung or peritoneum metastases) but these variables were not considered for subgroup analysis (Figures 2 and 3): please report these variables in subgroup analyses.

Author Response

Reviewer 2

This is a retrospective, multicenter (2 centers) study with the aim to investigate the efficacy and safety of two treatments (FTD-TPI+B vs FTD-TPI) as third or later-line treatment for refractory metastatic colorectal cancer patients. One hundred and six patients were selected between June 2020 and October 2024. The Authors find that patients treated with FTD-TPI+B had longer median progression-free survival and higher disease control rate compared to patients treated with FTD-TPI alone. Overall survival and safety were similar in the two treatments. The conclusion is that FTD-TPI+B should be considered for patients with good performance status (0-1). The study is interesting, however there are some points to clarify.

Comment 1: The main criticism is that this is a retrospective study, not randomized, thus the comparison between treatments should not be made. Moreover, the effectiveness should be considered instead of the efficacy.

Response 1: Thank you for your invaluable feedback. We fully agree with your point regarding the inherent limitations of a retrospective, non-randomized study design. As you have recommended, we explicitly addressed these limitations in the revised manuscript. Additionally, as this study was a retrospective observational analysis based on real-world clinical practice, we completely agree with your suggestion that the term "effectiveness" rather than "efficacy" is appropriate for our results. Accordingly, we have revised the manuscript, replacing "efficacy" with "effectiveness" to more accurately reflect the nature of our study. We greatly appreciate your insightful comments, which have significantly improved the clarity and quality of our manuscript.

The manuscript is revised as follows:

This study aims to assess the real-world effectiveness and safety of FTD–TPI+BEV compared to FTD–TPI alone as a third- or later-line treatment for refractory mCRC.

Effectiveness outcomes according to treatment regimen are summarized in Table 2.

Our study has several limitations. First, because the combination of bevacizumab with FTD–TPI was introduced as a treatment option following the SUNLIGHT trial results, the follow-up duration for patients treated with FTD–TPI plus bevacizumab was relatively short. Consequently, fewer death events occurred in this group, potentially limiting our ability to detect statistically significant differences in overall survival between treatment arms. Second, the relatively small patient cohort and the multiple statistical comparisons conducted increased the risk of type I errors; thus, statistically significant findings, especially those approaching significance thresholds, require careful interpretation. Moreover, the limited sample size reduced statistical power for subgroup analyses, resulting in wide confidence intervals and decreased reliability. Additionally, it precluded advanced statistical methods, such as propensity-score matching. Third, some patients experienced rapid clinical deterioration, precluding comprehensive radiologic assessment of treatment response and potentially confounding the evaluation of treatment effectiveness. Future studies with larger patient cohorts, allowing for thorough radiologic assessments and employing rigorous statistical methods, will be necessary to validate our findings and clarify treatment benefit.

Comment 2: A propensity-score matching samples should be performed, matching patients treated with the two treatments for some prognostic factors (see Austin PC. The use of propensity score methods with survival or time-to-event outcomes: reporting measures of effect similar to those used in randomized experiments. Stat Med 2014;33(7):1242-1258. doi: 10.1002/sim.5984)

Response 2: Thank you very much for this insightful comment. We fully agree that performing propensity-score matching (PSM) would significantly strengthen the comparability between the two treatment groups by adjusting for potential confounding factors. However, due to the limited sample size in our study, conducting PSM with multiple covariates would substantially reduce the number of matched patients, ultimately compromising statistical power and the reliability of our findings. We acknowledge this point as an important limitation of our study, and we will explicitly discuss this issue in the limitations section of the revised manuscript. Thank you again for highlighting this important methodological consideration.

The manuscript is revised as follows:

Our study has several limitations. First, because the combination of bevacizumab with FTD–TPI was introduced as a treatment option following the SUNLIGHT trial results, the follow-up duration for patients treated with FTD–TPI plus bevacizumab was relatively short. Consequently, fewer death events occurred in this group, potentially limiting our ability to detect statistically significant differences in overall survival between treatment arms. Second, the relatively small patient cohort and the multiple statistical comparisons conducted increased the risk of type I errors; thus, statistically significant findings, especially those approaching significance thresholds, require careful interpretation. Moreover, the limited sample size reduced statistical power for subgroup analyses, resulting in wide confidence intervals and decreased reliability. Additionally, it precluded advanced statistical methods, such as propensity-score matching. Third, some patients experienced rapid clinical deterioration, precluding comprehensive radiologic assessment of treatment response and potentially confounding the evaluation of treatment effectiveness. Future studies with larger patient cohorts, allowing for thorough radiologic assessments and employing rigorous statistical methods, will be necessary to validate our findings and clarify treatment benefit.

Comment 3: The NLR with a cut-off of 3 was used, but how this cut-off was chosen?

Response 3: We sincerely appreciate your thorough and insightful review. Regarding your comment on the choice of the NLR cut-off value, we used an NLR threshold of 3 based on previously published subgroup analyses from the RECOURSE trial, which compared the efficacy of FTD–TPI monotherapy versus placebo. In the RECOURSE trial, patients with low NLR (defined as NLR < 3) demonstrated improved PFS and disease control rate. We have specifically addressed this in our discussion section with the following statement: Consistent with findings from the RECOURSE trial [12], our study also identified a high NLR as an independent prognostic factor associated with inferior PFS and OS.”

Moreover, the SUNLIGHT trial similarly employed a subgroup analysis based on an NLR cut-off of 3. Although the prognostic significance of NLR is well established in oncology, the optimal cut-off value remains controversial. However, most meta-analyses assessing the prognostic role of NLR in various solid tumors have consistently reported a cut-off value around 3.0 (IQR 2.5–5.0) [Zahorec R et al., Bratisl Lek Listy. 2021;122(7):474-488]. Given these previous analyses, including those performed in both the RECOURSE and SUNLIGHT trials, we opted to adopt the same cut-off value of 3 in our study.

Comment 4: Please reports the numbers of patients at risk under the X-axis of each figure.

Response 4: We sincerely appreciate your thorough review. We fully agree with your suggestion regarding the necessity to include the number at risk in the Kaplan-Meier curves for PFS and OS. We have revised the figures accordingly. Thank you for pointing this out.

The manuscript is revised as follows:

Comment 5: In the statistical analysis, PFS and OS were defined but patients censored at the time of analysis were not reported. How the time of patients not progressed/death was considered?

Response 5: We are truly grateful for your thorough and thoughtful review. We fully agree with your comment regarding the need to clearly define censored patients, specifically those who did not experience disease progression or death by the time of analysis. In the Results section, we initially reported only the number of patients who experienced progression or death. In response to your suggestion, we will revise the manuscript to explicitly include the number of patients who were censored in both the PFS and OS analyses.

The manuscript is revised as follows:

OS was defined as the duration from chemotherapy initiation to death from any cause, and PFS was defined as the time from chemotherapy initiation to disease progression or death, whichever occurred first. Patients who did not experience disease progression or death by the data cutoff date were censored, with follow-up duration calculated from chemotherapy initiation to the date of last clinical follow-up or data cutoff.

The median follow-up duration was 5.9 months (95% CI, 4.4–7.3). Among the 106 patients, disease progression was observed in 96 (90.6%), with the remaining 10 patients (9.4%) censored at the time of analysis. For overall survival, 85 patients (80.2%) had died, and the remaining 21 patients (19.8%) were censored.

Comment 6: Please report univariate and multivariate analyses according to age in Table 3.

Response 6: We sincerely appreciate your valuable comment. In our initial analysis, we did not report the impact of age because the number of patients aged 65 years or older was relatively small, and age did not appear to have a significant association with prognosis in this cohort. However, we agree with your suggestion that reporting the analysis based on age would be informative. Therefore, we will add the results of the univariate analysis according to age in the revised manuscript. As for the multivariate analysis, since age was not identified as a significant factor in the univariate analysis, and including multiple non-significant variables in a multivariate model with a limited sample size could compromise statistical power, we chose not to include age in the multivariate model.

The manuscript is revised as follows:

PFS

OS

Variables

Univariate analysis

Multivariate analysis

Univariate analysis

Multivariate analysis

HR (95% CI)

p value

HR (95% CI)

p value

HR (95% CI)

p value

HR (95% CI)

p value

 Age ≥65 yr (vs. <65 yr)

0.84 (0.51–1.38)

0.498

1.11 (0.61–2.01)

0.734

Female (vs. male)

0.84 (0.56–1.27)

0.413

1.17 (0.76–1.80)

0.483

ECOG PS 2 (vs. PS 01)

2.11 (1.36–3.27)

0.001

1.56 (0.96–2.53)

0.071

3.30 (2.06–5.29)

<0.001

3.12 (1.88–5.19)

<0.001

Rectal (vs. Colon)

1.31 (0.87–1.98)

0.198

1.26 (0.81–1.96)

0.306

Right side (vs. Left side)

0.78 (0.44–1.38)

0.393

0.70 (0.36–1.36)

0.294

RAS Mutant (vs. Wild-type)

1.02 (0.67–1.56)

0.934

1.13 (0.72–1.77)

0.600

Previous lines of therapy >2 (vs. ≤2)

1.31 (0.84–2.06)

0.231

1.38 (0.88–2.15)

0.158

Metastatic duration (<18 vs. ≥18 months)

1.05 (0.67–1.66)

0.833

0.92 (0.56–1.51)

0.735

No. of sites of metastasis (≥3 vs. 0-2)

1.25 (0.83–1.88)

0.281

1.50 (0.96–2.34)

0.072

Liver metastases (vs. none)

1.44 (0.93–2.24)

0.105

0.96 (0.61–1.50)

0.844

Lung metastases (vs. none)

0.90 (0.57–1.40)

0.629

1.22 (0.75–1.99)

0.415

Peritoneum metastases (vs. none)

1.71 (1.12–2.61)

0.013

1.66 (1.07–2.58)

0.023

1.24 (0.80–1.92)

0.342

Neutrophil-lymphocyte ratio ≥3 (vs. <3)

1.63 (1.08–2.48)

0.022

1.60 (1.03–2.48)

0.038

1.79 (1.15–2.78)

0.010

1.64 (1.04–2.58)

0.032

CEA ≥50 μg/L (vs. <50 μg/L)

1.56 (1.03–2.37)

0.038

1.21 (0.77–1.91)

0.406

2.41 (1.50–3.87)

<0.001

1.93 (1.18–3.18)

0.009

FTD–TPI+BEV (vs. FTD–TPI)

0.56 (0.37–0.83)

0.004

0.60 (0.39–0.94)

0.025

0.74 (0.48–1.15)

0.189

1.09 (0.66–1.78)

0.742

Comment 7: In Table 3, the site of metastases was reported (liver, lung or peritoneum metastases) but these variables were not considered for subgroup analysis (Figures 2 and 3): please report these variables in subgroup analyses.

Response 7: Thank you very much for your insightful comment. We fully acknowledge your point regarding the clinical importance of metastatic sites as potential prognostic factors. As you noted, we did include these factors in our multivariable analysis (Table 3), recognizing their overall prognostic significance within the entire study cohort receiving FTD–TPI-based regimens. However, given that the main goal of our subgroup analyses was specifically to evaluate the comparative effectiveness between FTD–TPI+BEV and FTD–TPI monotherapy, we respectfully considered that additional subgroup analyses according to metastatic sites might not substantially contribute meaningful insights into the differential efficacy of these regimens and could inadvertently increase the risk of type I errors due to multiple statistical comparisons. Therefore, while fully appreciating your valuable suggestion, we have chosen not to incorporate metastatic sites into the subgroup analyses.

Round 2

Reviewer 1 Report

Comments and Suggestions for Authors

All issues were commented in detail and the manuscript revised in a sufficient way.

Reviewer 2 Report

Comments and Suggestions for Authors     Thanks to the Authors for answering all the questions